# Molecular basis of human trace amine-associated receptor 1 activation

Gregory Zilberg [1] ✉, Alexandra K. Parpounas [2], Audrey L. Warren [2], Shifan Yang [3] & Daniel Wacker [1,2,3] ✉

The human trace amine-associated receptor 1 (hTAAR1, hTA1) is a key regulator of monoaminergic neurotransmission and the actions of psychostimulants. Despite preclinical research demonstrating its tractability as a drug target, its molecular mechanisms of activation remain unclear. Moreover, poorly understood pharmacological differences between rodent and human TA1 complicate the translation of findings from preclinical disease models into novel pharmacotherapies. To elucidate hTA1's mechanisms on the molecular scale and investigate the underpinnings of its divergent pharmacology from rodent orthologs, we herein report the structure of the human TA1 receptor in complex with a Gαs heterotrimer. Our structure reveals shared structural elements with other TAARs, as well as with its closest monoaminergic orthologue, the serotonin receptor 5-HT4R. We further find that a single mutation dramatically shifts the selectivity of hTA1 towards that of its rodent orthologues, and report on the effects of substituting residues to those found in serotonin and dopamine receptors. Strikingly, we also discover that the atypical antipsychotic medication and pan-monoaminergic antagonist asenapine potently and efficaciously activates hTA1. Together our studies provide detailed insight into hTA1 structure and function, contrast its molecular pharmacology with that of related receptors, and uncover off-target activities of monoaminergic drugs at hTA1.

The human trace amine (-associated) receptor 1 (hTAAR1, hTA1) has emerged in the past 15 years as a key modulator in monoaminergic neurotransmission as a rheostatic feedback mechanism[1]. Shortly after its initial cloning and confirmation as a high-affinity receptor for the trace amines β-phenethylamine (β-PEA) and tyramine (TYR), reports of μM potency of amphetamine and methamphetamine at TA1, as well as its localization in several monoaminergic nuclei, suggested that it may play a key role in mediating the effects of amphetamine-type psychostimulants[2,3]. Subsequently, it was established that hTA1 signaling modulates the membrane localization of mono-aminergic transporters[4], and its activation suppresses spontaneous dopaminergic neuron firing[5]. Notably, this receptor appears to localize primarily intracellularly and may couple to different downstream effectors in different organelles[6].

hTA1 has been highlighted as a potential target for treating disorders of dopaminergic dysfunction, such as schizophrenia and methamphetamine use disorder[7,8], as well as metabolic disorders, cognitive impairments, and sleep-related dysfunction[7,9,10]. Several pharmaceutical organizations have thus conducted drug discovery campaigns to develop hTA1-focused pharmacotherapies. Initial clinical development efforts by Roche were hampered by pharmacokinetic issues[11]. Studies have also implied that the translation of preclinical studies to clinical applications has been impaired by strongly divergent pharmacological properties of human and rodent TA1[12]. Nonetheless,

[1]Department of Neuroscience, Icahn School of Medicine at Mount Sinai, New York, NY 10029, USA. [2]Department of Pharmacological Sciences, Icahn School of Medicine at Mount Sinai, New York, NY 10029, USA. [3]Department of Genetics and Genomic Sciences, Icahn School of Medicine at Mount Sinai, New York, NY 10029, USA. ✉e-mail: greg.zilberg@icahn.mssm.edu; daniel.wacker@mssm.edu

the hTA1/5-HT1AR agonist ulotaront (SEP-363856)[13] has received FDA Breakthrough status. The compound showed promising results in recent Phase 2 trials for Parkinson's disease psychosis[14], as well as the treatment of schizophrenia[15], supporting the notion that hTA1 may be a tractable therapeutic target[16]. While ulotaront has failed to meet primary endpoints in two recent Phase 3 trials for schizophrenia[17,18], additional Phase 2/3 trials are scheduled to test its efficacy in the treatment of anxiety and depression[19,20]. Despite these clinical developments, the molecular mechanisms by which hTA1 transduces signals, however, remain poorly understood, especially in comparison to the better-studied receptors for serotonin, dopamine, histamine, acetylcholine, and epinephrine, with which it shares considerable homology.

Here we report the cryoEM structure of a hTA1·Gs signaling complex to illuminate hTA1's molecular mechanisms and elucidate similarities and differences with other TAAR members and rodent TA1 receptors on the atomic scale. We further interrogate several modulatory surfaces including hTA1's presumed orthosteric binding pocket via in vitro pharmacological assays and mutational studies. Lastly, due to the observed structural similarity to monoaminergic receptors, we also screen a small library of aminergic drugs and research compounds, and identify and characterize the atypical antipsychotic asenapine as a potent and efficacious hTA1 agonist.

## Results

### Pharmacological characterization of known hTA1 ligands

We first measured the activity of a panel of known hTA1 agonists to validate previous data and identify a suitable ligand for structural studies. To determine compound-mediated hTA1 activation, we measured Gs-mediated increases in cellular cAMP levels in

HEK293T cells using a cAMP biosensor[21] (Fig. 1A). We chose a small compound panel and determined activities relative to the endogenous agonist β-PEA ($EC_{50} = 80.6$ nM, designated $E_{max} = 100.0\%$). All compound activities in this manuscript are reported as $EC_{50}$ and $E_{max}$, and standard errors as well as $pEC_{50}$ values can be found in Supplementary Table 1. The tested compounds include the endogenous agonists TYR ($EC_{50} = 414.9$ nM, $E_{max} = 99.0\%$ of β-PEA) and 3-iodothyronamine ($T_1AM$; $EC_{50} = 742.6$ nM, $E_{max} = 70.5\%$), the preclinical compounds Ro5256390 ($EC_{50} = 5.3$ nM, $E_{max} = 103.3\%$) and Ro5263397 ($EC_{50} = 1.48$ nM, $E_{max} = 86.7\%$), and the clinical candidates ulotaront ($EC_{50} = 180.0$ nM, $E_{max} = 109.07\%$) and ralmitaront ($EC_{50} = 110.4$ nM, $E_{max} = 40.1\%$) (Supplementary Table 1). Ro5256390, a previously developed hTA1-selective preclinical candidate with efficacy in rodent addiction models, showed comparable efficacy to that of β-PEA (Fig. 1A). Moreover, Ro5256390 showed a potency of ~5 nM in our assay, validating previous findings that it is currently one of the most potent and efficacious hTA1 agonists. These results also suggested the utility of Ro5256390 in forming stable hTA1·Gs complexes for structural studies of hTA1's molecular mechanisms and pharmacology.

### Determination of a Ro5256390-bound hTA1·Gs structure

To elucidate the molecular architecture of hTA1 and provide insight into the receptor's activation mechanisms and drug binding surfaces, we determined a cryoEM structure of Ro5256390-bound hTA1 in complex with a heterotrimeric Gs protein (Fig. 1B, Supplementary Fig. 1). To this end, we expressed and purified hTA1 from *Sf9* cells using a full-length hTA1 construct bearing the stabilizing[22] mutation F112$^{3.41}$W (Superscripts denote Ballesteros-Weinstein numbering[23]), as well as an N-terminal b562RIL apocytochrome (BRIL)[24] fusion followed by the first

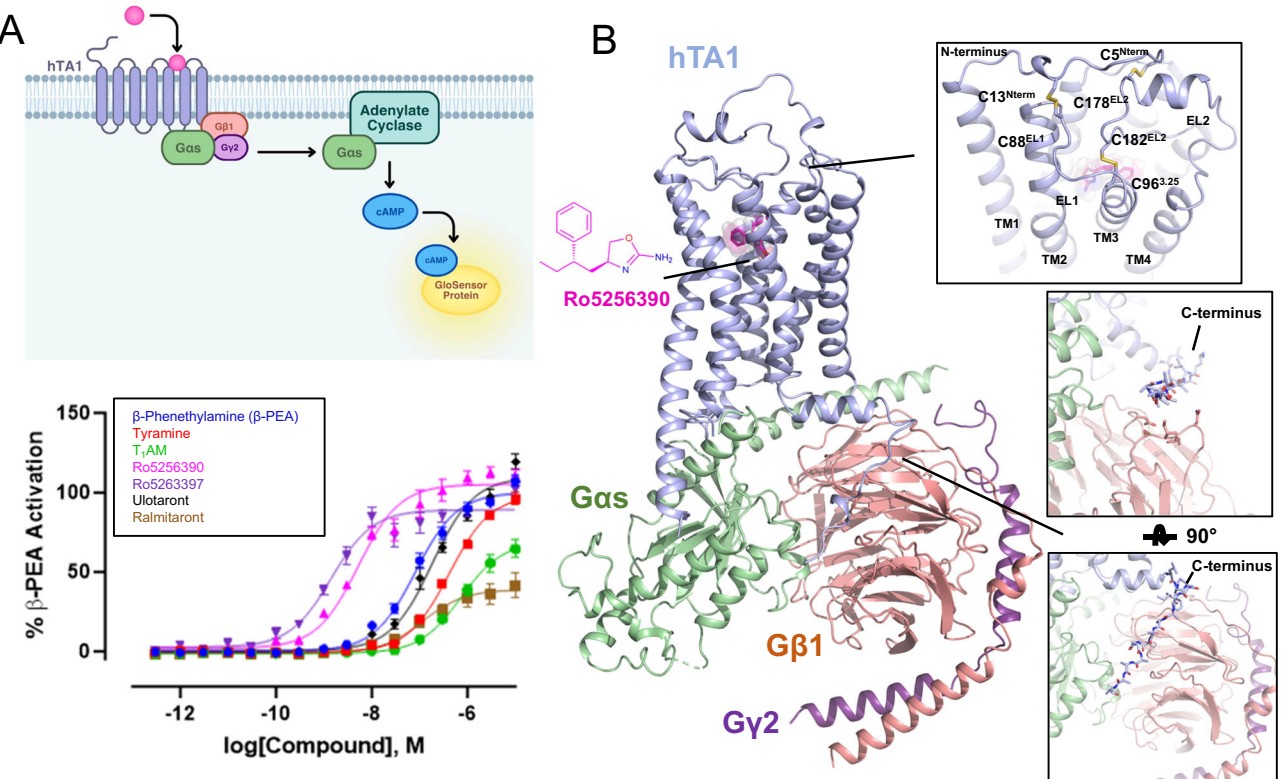

**Fig. 1 | Pharmacological and structural characterization of the hTA1·Gs complex. A** Schematic diagram of the GloSensor cAMP accumulation assay used in this study (top) and hTA1 activation in transfected HEK293T cells mediated by a panel of agonists (bottom), data represent mean ± SEM of three independent experiments (n = 3) performed in triplicate. See Supplementary Table 1 for fitted parameter values. **B** Cartoon view of the Ro5256390-bound hTA1·Gs complex cryo-EM structure. Nb35 has been removed for clarity. Zoom-ins highlight extracellular surfaces with disulfide bridges and views of the receptor C-terminus. Light blue, hTA1; Ro5256390, magenta; Gαs, green; Gβ1, salmon; Gγ2, purple.

25 residues of the human β2-adrenergic receptor (β2AR) to increase hTA1 expression yields[25]. This construct only shows minor differences when compared to the wildtype receptor in signaling studies (Supplementary Fig. 1, Supplementary Table 1), most of which, we surmise, are due to increased receptor trafficking to the plasma membrane[25]. Following our previously established protocols, we obtained heterotrimeric Gs via co-expressing Gβ1 and a Gγ2-Gαs fusion, using previously reported stabilized dominant negative mutations (see methods) in Gαs[26]. Ro5256390-bound receptor and Gs were purified separately, and complexes were formed overnight in the presence of Nb35, which stabilizes the Gs heterotrimer[27] (Supplementary Fig. 1).

## Overall complex architecture

We obtained an initial cryoEM structure of Ro5256390-bound hTA1-Gs-Nb35 at a global nominal resolution of 3.35 Å, with local resolutions as high as 3.0 Å around the ligand binding pocket (Supplementary Fig. 1). At this resolution, we were able to unambiguously place the compound and elucidate the ligand-binding pocket of hTA1 (Supplementary Fig. 2A, B). Overall, the structure of the complex is reminiscent of those previously reported for other GPCR-G protein complexes, where the C-terminal helix of the Gα subunit extends into the receptor transducer binding site formed largely by the cytoplasmic ends of transmembrane helices (TM) 2, 3, 5, 6, and 7[27]. However, we do observe considerable flexibility in several areas of the complex, including hTA1's extracellular surface and C-terminus, as well as parts of the receptor-G protein interface (Supplementary Fig. 2).

For instance, we observed continuous and clear density for the receptor C-terminus in unsharpened cryoEM maps (Fig. 1B, Supplementary Fig. 2C). Accordingly, the hTA1 C-terminus interacts with Gβ, but the low resolution of this area (several attempted data processing approaches failed to improve resolution) precludes both identification of sidechain rotamer states in the C-terminus as well as detailed interactions between hTA1 and Gβ. We thus modeled a poly-alanine chain into the low-resolution density. We were, however, able to characterize the receptor's extracellular surface following local refinement and 3D variability analysis that supported the modeling of several key extracellular features. First, part of extracellular loop 2 (EL2) forms a 2-turn helix, which stacks atop the N-terminal end of EL2 (Supplementary Fig. 2D). This loop originates from TM5 and stretches across the 7 transmembrane helix (7TM) bundle, forming the boundary of the ligand binding pocket (Fig. 1B). Second, we observed that the N-terminus of hTA1 folds over the receptor, and forms two disulfide bridges with EL1 and EL2, distinct from the single disulfide bridge between the N-terminus and EL2 found in the structure of mouse TAAR9 (mTAAR9)[28] (Supplementary Fig. 3). 3D variability analysis suggests two stable points of density extending from EL1 and EL2 that connect more variable density that corresponds to the extension of the N-terminus emerging from TM1 (Supplementary Movie 1). However, mutating C88$^{EL1}$ to a serine had virtually no effect on compound potencies, while mutating C5$^{Nterm}$, C13$^{Nterm}$, and C178$^{EL2}$ moderately reduced compound potencies (Supplementary Fig. 3, Supplementary Table 1), with lesser effects observed for TYR and β-PEA at all but C13$^{Nterm}$S. Conversely, mutating either C88$^{EL1}$ or C178$^{EL2}$ to a serine appeared to only moderately increase the efficacies of Ro5256390 (E$_{max}$=121.7% and 119.6%), Ro5263397 (E$_{max}$=100.3% and 105.2%), and ralmitaront (E$_{max}$=77.9% and 69.5%) relative to β-PEA (Supplementary Fig. 3, Supplementary Table 1). We considered the possibility that these disulfides benefit from potential redundancy with each other, as well as with the conserved disulfide between EL2 and TM3, which is observed in the vast majority of Class A GPCRs[29]. However, even a C88$^{EL1}$S/C178$^{EL2}$S double mutant did not dramatically impact hTA1 ligand potency (at most a 2.4-fold reduction for ulotaront), suggesting that both N-terminal disulfide bridges are largely dispensable to signaling mediated by β-PEA and other tested agonists. By comparison, mutation of the conserved C96$^{3.25}$ residue to a serine ablated dose-dependent cAMP accumulation, indicating that these disulfides do not offer compensatory stabilization of the conserved EL2-transmembrane core interface (Supplementary Fig. 3).

3D variability analysis further uncovered considerable motion at the G protein-receptor interface by showing variability in TM5, TM6, and the angle by which Gαs' C-terminal helix engages the receptor. We observe TM1 moving between H8 and TM2 (Supplementary Movie 2), and a corresponding motion of TM4 between TM2 and TM5 (Supplementary Movie 3). These motions occur in concert with a rotating motion and simultaneous disordering of the cytoplasmic ends of TM5 and TM6 (Supplementary Movie 4). In total, this results in a slight twisting and rocking motion around the C-terminal helix of Gαs (Supplementary Movie 5).

## hTA1 ligand binding site

Ro5256390 is bound to hTA1 in a pocket near the extracellular region of the receptor, commonly termed the orthosteric binding pocket (OBP). As is conserved in aminergic GPCRs, the compound interacts with the conserved aspartate D103$^{3.32}$, which is positioned near the amine-substituted oxazoline moiety (Fig. 2A). The phenyl moiety of the compound extends towards TM5, and is stabilized by hydrophobic interactions with F267$^{6.51}$ and F268$^{6.52}$ in TM6, I104$^{3.33}$ in TM3, as well as F186$^{EL2}$ and V184$^{EL2}$ in EL2. Overall, the binding mode of the phenyl moiety is similar to that of β-PEA bound to mTAAR9 (Supplementary Fig. 4). Strikingly, aside from D$^{3.32}$ and Y$^{7.43}$, none of the residues in the binding pockets of hTA1 and mTAAR9 are conserved. It is worth noting that F$^{6.51}$ and F$^{6.52}$ have been suggested to stabilize the aromatic core moieties of serotonin, dopamine, and norepinephrine, and are universally conserved across their respective receptors[30]. The finding that hTA1 is the only subtype of the human TAAR family that contains both phenylalanines, therefore, further suggests a closer relationship with other neurotransmitter receptors than other trace amine receptors.

## General Determinants of hTA1 ligand binding

As expected, we found that mutation of the conserved amine-coordinating D103$^{3.32}$ to asparagine all but abolished dose-dependent responses to all ligands tested (Supplementary Table 1). In our structure, Ro5256390's amino-oxazoline ring extends towards the 7TM core into a crevice formed by L72$^{2.53}$, W264$^{6.48}$, Y294$^{7.43}$, and S107$^{3.36}$. We probed this interaction by mutating S107$^{3.36}$ to cysteine, a common residue at that position in many monoaminergic receptors, and observed that the potency of Ro5256390 (EC$_{50}$ = 504.4 nM) and Ro5263397 (EC$_{50}$ = 247.8 nM) decreased by approximately 100-fold. By comparison, the potencies of β-PEA (EC$_{50}$ = 205.4 nM) and TYR (EC$_{50}$ = 186.4 nM) did not substantially change (Fig. 2, Supplementary Fig. 4, Supplementary Table 1). These findings highlight that a bulkier and nonpolar cysteine sidechain disproportionally affects the binding of sterically demanding amino-oxazoline compounds. Analogously, previous computational work has implicated hydrogen bonding between ulotaront and S107$^{3.36}$ as being key for its potent binding[31]. Concordant with this hypothesized interaction, we observe a nearly 10-fold decrease in ulotaront potency at the S107$^{3.36}$C mutant (EC$_{50}$ = 1668.0 nM), albeit with an increased efficacy relative to β-PEA (E$_{max}$=197.2%) (Fig. 2, Supplementary Table 1).

We further mutated the conserved toggle switch W264$^{6.48}$, which strongly and indiscriminately decreased the potency of all ligands tested by at least 10-fold, notably decreasing the potency of Ro5263397 by more than 1000-fold (EC$_{50}$ = 5241.1 nM). Lastly, we mutated the residue R83$^{2.64}$, as this arginine is conserved across all hTAARs and extends into the periphery of the binding pocket. A R83$^{2.64}$H mutation dramatically affected the activities of most tested compounds to a point where potencies could no longer be accurately determined for TYR, T$_1$AM, and ralmitaront (Fig. 2B). While R83$^{2.64}$H did not appear to affect potencies of Ro5256390 and Ro5263397, it strongly reduced their efficacies.

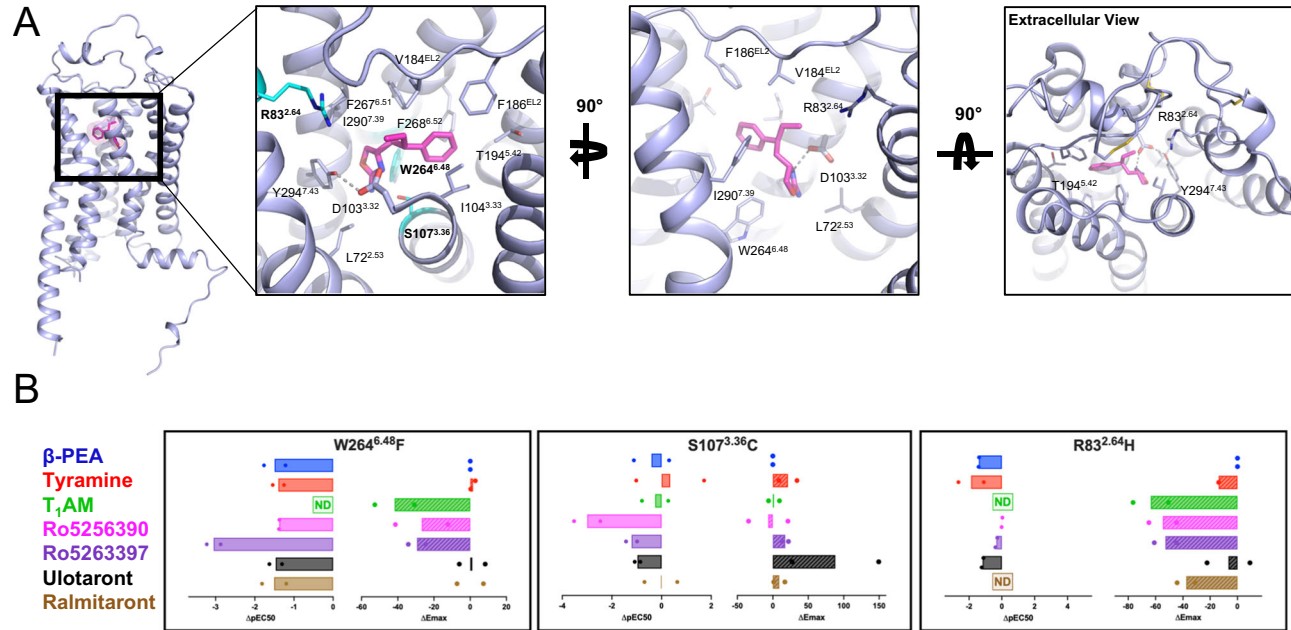

**Fig. 2 | Features of the Ro5256390-bound orthosteric binding pocket of hTA1.**
**A** The orthosteric binding pocket (OBP) of hTA1 and key residues are shown from three angles, with residues chosen for mutagenic study highlighted in cyan. Grey dashed lines denote an ionic interaction between Ro5256390 and the receptor. Light blue, hTA1; Ro5256390, magenta. **B** Effects of distinct mutations of the potency and efficacy of hTA1 agonists. Data are shown as differences between mutant and wild type receptor with potencies determined as $pEC_{50}$s and efficacies measured as $E_{max}$. Data represent mean of two independent experiments ($n = 2$). ND denotes that no reliable $pEC50/\Delta pEC50$ values could be obtained due to low compound activity. See Supplementary Table 1 for fitted parameter values.

## Structural similarity to other TA1s

Studies have shown that there are considerable functional and pharmacological differences between hTA1 and TA1 in rats (rTA1) or mice (mTA1)[12], with key implications for translating findings from preclinical models into human therapies. For instance, TYR has been reported to be ~30 times more potent at rTA1 than hTA1, and the antagonist EPPTB was shown to have an affinity of ~1 nM at mTA1 but does not appear to bind hTA1[32]. Inversely, the recently reported TA1 antagonist RTI-7470-44 has an $IC_{50}$ of about 8 nM at hTA1 but shows ~90-fold and ~140-fold reduced potencies at rTA1 and mTA1, respectively[33]. These species-level differences already impact the signaling of endogenous ligands. For instance, the thyroid hormone derivative $T_1AM$ is a nanomolar affinity full agonist at rTA1 (and to a lesser extent also at mTA1) but a weaker partial agonist at hTA1[34], and tryptamine shows over 50-fold higher potency at rTA1 compared to hTA1[12]. Similarly, these considerable differences also impact the TA1 activities of drugs, as several potent psychedelics reportedly have more than 1000-fold selectivity for rTA1 over hTA1[12]. Sequence analysis reveals that hTA1 exhibits 83% and 82% similarity to rTA1 and mTA1, respectively, and our structural comparison shows that species differences are found in virtually all helices and domains of hTA1 (Fig. 3A, Supplementary Fig. 5). Although this diversity can potentially also account for divergent receptor trafficking, transducer binding, and allosteric coupling between ligand and transducer binding sites, we wanted to investigate the impact of ligand binding site differences on the activities of hTA1 agonists.

Out of the residues comprising the ligand binding pocket of Ro5256390, three are different in mTA1, and four are different in rTA1 (Fig. 3A). Perhaps the most apparent differences between human and rodent TA1 are found at positions 7.39, 5.42, and 45.52 (EL2). These residues have long been known to form direct ligand interactions in other aminergic receptors[35]. We thus set out to test how substituting ligand-binding pocket residues in hTA1 with those of corresponding residues found in rodent TA1s would affect the activities of TA1 ligands.

First, $I290^{7.39}$, which is a tyrosine in mTA1 and an asparagine in rTA1, was previously identified as a key factor responsible for species differences in responsiveness to $T_1AM$[36], although this work did not assess hTA1 residues. When we mutated $I290^{7.39}$ to a tyrosine as in mTA1, we observed increased potencies for Ro5256390 ($EC_{50} = 3.8$ nM), Ro5263397 ($EC_{50} = 0.7$ nM), and particularly ulotaront ($EC_{50} = 30.3$ nM), while the potencies of β-PEA ($EC_{50} = 371.9$ nM) and TYR ($EC_{50} = 1962.7$ nM) decreased (Fig. 3B, Supplementary Fig. 5, Supplementary Table 1). Conversely, mutation of $I290^{7.39}$ to an asparagine as in rTA1 further increased the potency of ulotaront ($EC_{50} = 1.4$ nM) by over 100-fold. $T_1AM$'s potency ($EC_{50} = 139.3$ nM) increased by about 5-fold, whereas the potencies of β-PEA ($EC_{50} = 54.7$ nM) and TYR ($EC_{50} = 497.1$ nM) were largely unchanged.

$T194^{5.42}$ is another residue specific to hTA1, as mTA1 and rTA1 both contain an alanine at this position. This is a particularly drastic difference, as the polar residues found at position 5.42 in dopamine and norepinephrine receptors have been implicated in directly interfacing with one of the hydroxyl groups in dopamine[37] and norepinephrine[38]. Unexpectedly, a $T194^{5.42}A$ mutation increases TYR's potency by over 10-fold (Fig. 3B). This suggests that TYR does not form a hydrogen bond with $T194^{5.42}$ that contributes to its potency and instead benefits from a substitution to a sterically smaller alanine as found in rodent TA1s. By contrast, $T194^{5.42}A$ reduces the potencies of Ro5263397 ($EC_{50} = 56.4$ nM) and Ro5256390 ($EC_{50} = 34.8$ nM) by over 30-fold and over 5-fold, respectively, which is surprising given that Ro5256390 is located ~5 Å from the $T194^{5.42}$ sidechain, and Ro5263397's smaller size suggests an even greater distance. The $T194^{5.42}A$ mutation also increases the efficacy of all compounds relative to β-PEA's, with the exception of ralmitaront, with the most pronounced increase for ulotaront ($E_{max}=179.7\%$ of β-PEA). It should be noted, however, that ulotaront's potency ($EC_{50} = 1030.06$ nM) decreases nearly 8-fold (Fig. 3B, Supplementary Fig. 5, Supplementary Table 1).

Lastly, $V184^{45.52}$ in EL2, which is a key residue in aminergic receptors that has been implicated in ligand kinetics[39,40] and biased signaling[40,41], is a proline in both mTA1 and rTA1. Our structure reveals direct hydrophobic contacts between $V184^{45.52}$ and Ro5256390, and we anticipated that substitution with proline would likely affect ligand

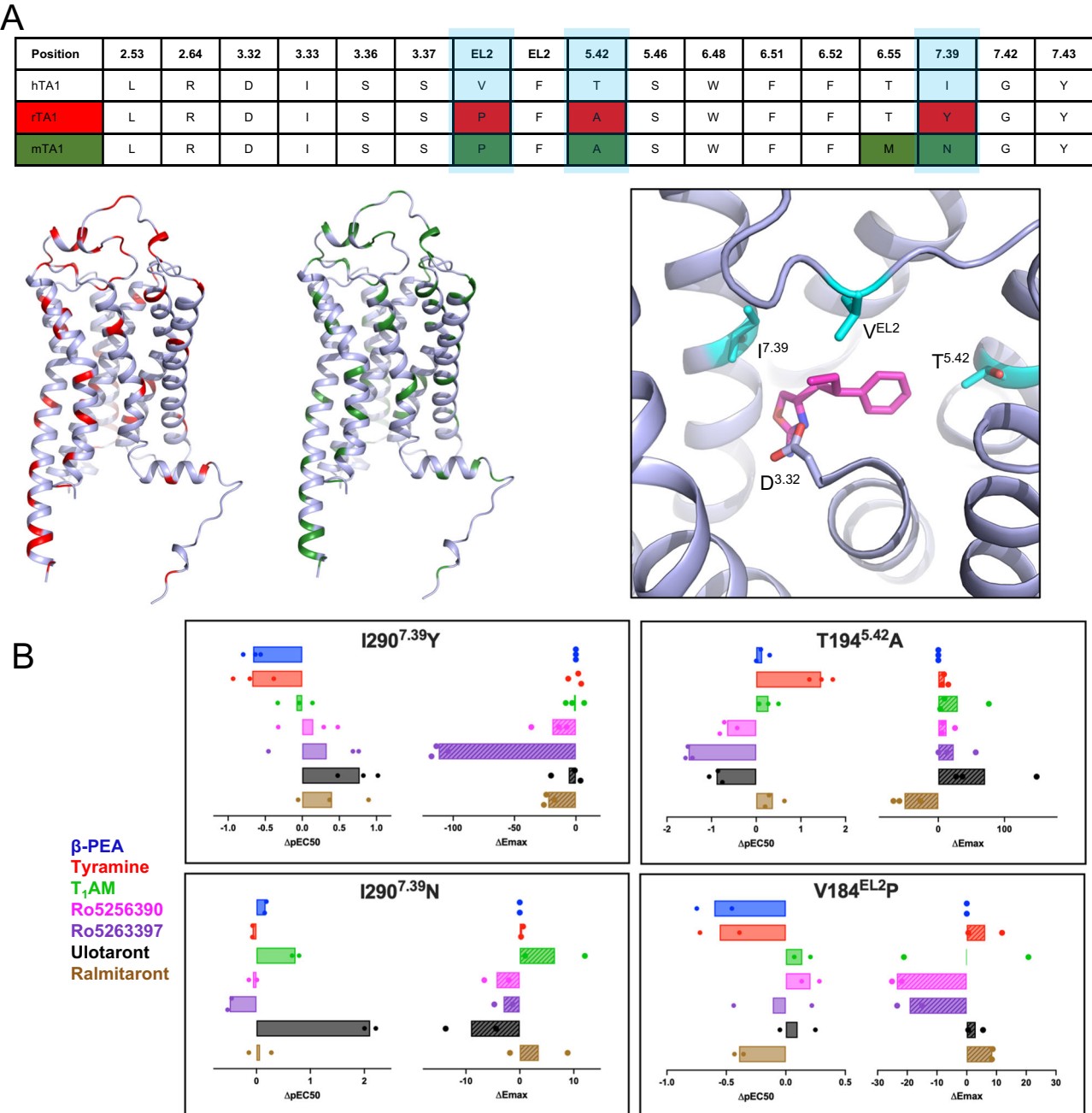

**Fig. 3 | Species differences between human and rodent TA1. A** Sequence alignment of residues in conserved positions within the OBP of TA1 (top), with highlighted differences between hTA1 and rTA1 (red) or mTA1 (green). Locations of differences with rTA1 (red) or mTA1 (green) are further highlighted on the structure of hTA1. Different residues in the OBP of hTA1, rTA1, and mTA1 are shown in cyan. $D^{3.32}$ is highlighted for spatial reference and Ro5256390 is shown in magenta. **B** Effects of species-related mutations in the OBP on the potency and efficacy hTA1 agonists. Data are expressed as differences between mutant and wild type receptor with potencies determined as $pEC_{50}$s and efficacies measured as $E_{max}$. Data represent mean of two ($n = 2$; $I290^{7.39}N$, $V184^{EL2}P$) or three ($n = 3$; $I290^{7.39}Y$, $T194^{5.42}A$) independent experiments. See Supplementary Table 1 for fitted parameter values.

interactions and change the positioning of the EL2 backbone due to the structural constraints of this residue. Interestingly, we find that while a $V184^{45.52}P$ mutation does reduce the potencies of β-PEA ($EC_{50} = 321.2$ nM), TYR ($EC_{50} = 1490.2$ nM), and ralmitaront ($EC_{50} = 273.0$ nM) by 3-, 4-, and 2.5-fold, respectively, it does not appear to affect the potencies of Ro5256390 ($EC_{50} = 3.3$ nM) or the other tested compounds.

Overall, our findings provide detailed molecular insight into how even single amino acid differences in the binding pockets of human and rodent TA1 can have drastic effects on the activities of endogenous ligands and clinical candidate drugs such as ulotaront. Our studies thereby further emphasize how TA1-related pharmacology observed in preclinical rodent models needs to be carefully evaluated in the context of marked species differences.

## Structural similarity to neurotransmitter receptors

TAARs as a group are frequently categorized as olfactory receptors due to their role in odorant perception[42,43], while hTA1, which is not expressed in the olfactory epithelium[44], is often mentioned alongside neurotransmitter receptors such as serotonin and dopamine receptors[45]. We thus performed structure and sequence analysis to uncover similarities with aminergic neurotransmitter receptors

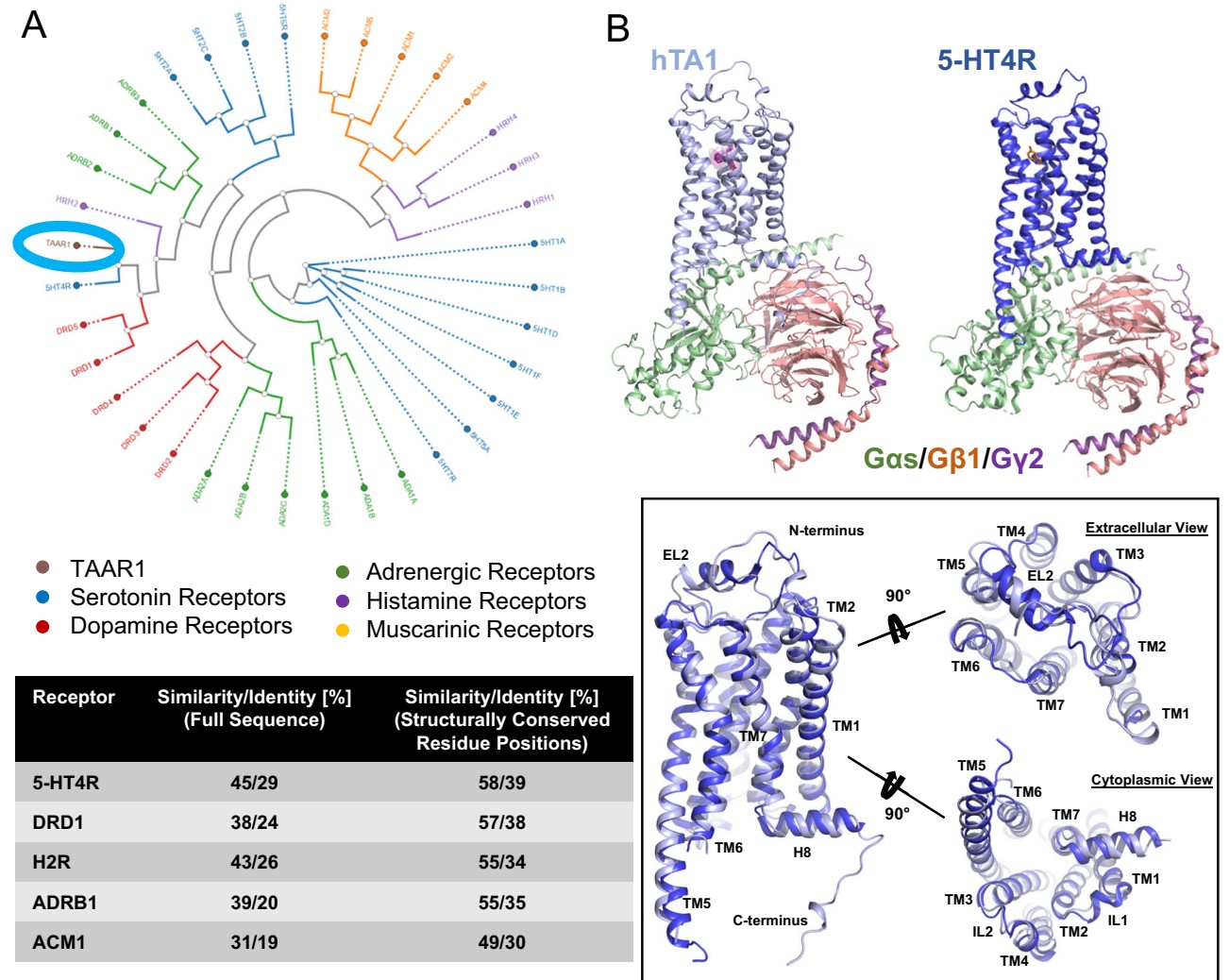

| Receptor | Similarity/Identity [%] (Full Sequence) | Similarity/Identity [%] (Structurally Conserved Residue Positions) |
|---|---|---|
| 5-HT4R | 45/29 | 58/39 |
| DRD1 | 38/24 | 57/38 |
| H2R | 43/26 | 55/34 |
| ADRB1 | 39/20 | 55/35 |
| ACM1 | 31/19 | 49/30 |

**Fig. 4 | Structural similarity to aminergic neurotransmitter receptors.**
**A** Phylogenetic analysis of the structurally conserved residues of human aminergic receptors using the GPCRdb tool (Top). hTA1 is circled for orientation. Sequence similarity and identity calculated for complete receptor sequences or only structurally conserved positions (bottom). **B** Structural comparison of the hTA1-Gs and 5-HT4R-Gs (PDB: 7XT8) complexes and superposition of the receptors shown from different angles. Light blue, hTA1; dark blue, 5-HT4R; Ro5256390, magenta; orange, serotonin; Gαs, green; Gβ1, salmon; Gγ2, purple.

(Fig. 4A). Phylogenetic analysis using the GPCRdb tool[46] shows hTA1's high sequence similarity and identity with serotonin, dopamine, histamine, adrenaline, and – to a lesser extent – muscarinic acetylcholine receptors (Fig. 4A). This analysis identifies the 5-HT4 serotonin receptor (5-HT4R) and D1 dopamine receptor (DRD1) as the closest related aminergic GPCRs. hTA1 exhibits 45%/29% sequence similarity/identity with 5-HT4R, and even 58%/39% sequence similarity/identity if only structurally conserved residue positions are considered. Strikingly, these percentages are similar to those obtained when hTA1 is compared to other human TAARs, and hTA1's structurally conserved regions appear as similar to 5-HT4R's as they are to hTAAR8's (Supplementary Fig. 6).

Having determined the structure of hTA1, we next wanted to investigate whether this observed sequence similarity with other aminergic receptors extends to structural similarity. hTA1's structure appears most similar to that of 5-HT4R, validating our findings from sequence analysis (Fig. 4B). As is the case for the herein reported hTA1-Gs complex structure, 5-HT4R's structure was determined in complex with a Gs heterotrimer[47], facilitating structural comparison. Structural alignment shows an RMSD of 0.613 Å when both receptor-G protein complex structures are compared, and 0.941 Å when only the receptors are aligned. As observed for hTA1, 5-HT4R features a 2-turn helix

located above the N-terminus of EL2, although the functional significance of this structural motif remains unidentified.

Regarding the OBP, the 17 residues that comprise the Ro5256390 binding pocket of hTA1 show higher similarity to 5-HT2Rs, 5-HT7R, as well as DRD1 and DRD5 dopamine receptors than to other hTAARs (Supplementary Fig. 6). hTA1's reduced affinity for monoaminergic neurotransmitters[2,3] is thus less likely solely due to the lack of direct contacts within the ligand binding pocket. We hypothesized instead that subtle remodeling of the ligand binding pocket, as well as greater changes in the surrounding residue environment, might alter key transition states that would otherwise enable high affinity neurotransmitter binding. We therefore interrogated compound interactions with TM5 and TM6 residues, which are critical for activation of aminergic receptors.

We first noted that a major difference in the proximity of the hTA1 OBP compared to other aminergic receptors is how F195[5.43] orients in the space between TM5 and TM6 and interacts with F268[6.51] (Fig. 5A). Previous studies focusing on ulotaront's binding at hTA1 have further suggested that it directly participates in aromatic π-π edge-to-face interactions with F195[5.43] [31,48]. We note that the rotameric state of F195[5.43] observed in our density precludes direct ligand-binding interactions, and that mutation to threonine marginally

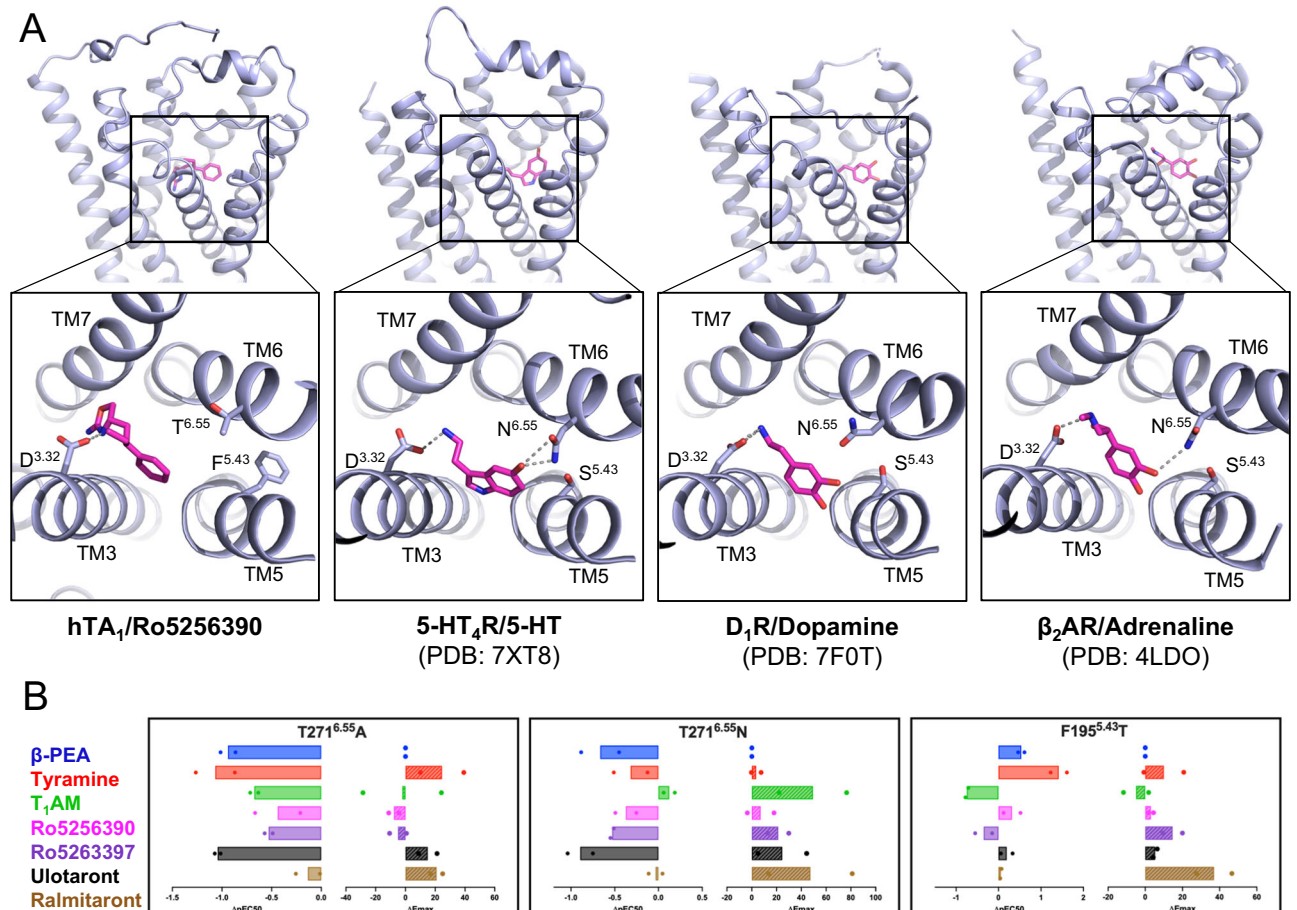

**Fig. 5 | Binding pocket similarities between hTA1 and aminergic neurotransmitter receptors. A** Side-by-side comparison of the Ro5256390-bound hTA1 OBP, the serotonin-bound 5-HT4R OBP (PDB: 7XT8), the dopamine-bound DRD1 OBP (PDB: 7F0T), and the norepinephrine-bound β2AR OBP (PDB: 4LDO). Key residues are shown as sticks, grey dashed lines represent ionic bonds between the conserved $D^{3.32}$ residue and the aminergic ligands. **B** Effects of OBP mutations on the activities of hTA1 agonists. Data are expressed as differences between mutant and wild type receptor with potencies determined as $pEC_{50}$s and efficacies measured as $E_{max}$. Data represent mean of two independent experiments ($n = 2$). See Supplementary Table 1 for fitted parameter values.

increases ulotaront's potency and efficacy ($EC_{50} = 114.9$ nM; $E_{max} = 114.4\%$), suggesting that a putative interaction with this residue is dispensable.

Previous reports suggested that a conserved pair of polar residues in dopamine and serotonin receptors located at 5.43 and 6.55 form an interhelical hydrogen bond that may be key for high-affinity recognition of the hydroxyl group of their respective ligands[49]. Indeed, mutation of $F195^{5.43}$ to a threonine increased the potencies of TYR ($EC_{50} = 15.8$ nM) and β-PEA ($EC_{50} = 23.5$ nM), representing a ~20-fold increase for TYR and a ~3.5-fold increase for β-PEA (Fig. 5B, Supplementary Table 1). The stronger increase in TYR's potency compared to β-PEA's conceivably further supports the importance of this polar interaction in coordinating ligand hydroxyl groups oriented toward TM6. Conversely, mutating $T271^{6.55}$ to an alanine reduces the potency of all compounds tested, with both β-PEA's ($EC_{50} = 701.1$ nM) and TYR's ($EC_{50} = 4846.8$ nM) potency decreasing by about 10-fold (Fig. 5B). Mutating $T271^{6.55}$ to an asparagine, on the other hand, disproportionately affected the potency of β-PEA ($EC_{50} = 374.4$ nM) relative to TYR ($EC_{50} = 862.9$ nM). Interestingly, the $T271^{6.55}$N mutation increased the efficacy of both $T_1AM$ ($E_{max}=119.8\%$) and ralmitaront ($E_{max}= 87.4\%$) relative to β-PEA, with negligible effects on compound potency (Fig. 5B, Supplementary Table 1). It is thus conceivable that $T_1AM$ and ralmitaront form hydrogen bonds with the $T271^{6.55}$N side-chain, as this structurally conserved residue has been shown to form similar drug interactions in other receptors[50].

## Discovery of asenapine as a potent hTA1 agonist
Due to hTA1's structural similarity to other aminergic receptors such as 5-HT4R, especially in its OBP, we reasoned that known aminergic drugs and research compounds likely have undiscovered off-target activities at hTA1. To uncover such activities and provide further insight into the receptor's structural and pharmacological similarity to other neurotransmitter receptors, we screened a library of 89 select aminergic compounds (Fig. 6A, Supplementary Table 3). cAMP accumulation studies using 10 μM of compound revealed hTA1 activation by 16 compounds, defined by at least a 4-fold (log2-fold change of 2) increase of signal over DMSO-treated controls (Supplementary Table 3). These compounds included the monoamines serotonin, epinephrine, and histamine, which have all previously been reported to be weak μM-potency compounds at hTA1[2], as well as the positive control β-PEA. Additionally, we observed activity for the ergoline compounds ergotamine, lisuride, LSD, and pergolide, which have previously been reported to be agonists of the rTA1 orthologue, although their potencies have not been reported[3]. Unexpectedly, we further observed hTA1 activation by quinpirole, quipazine, oxymetazoline, WAY161503, Ro600175, lorcaserin, and asenapine. To validate these findings, we next performed concentration response experiments in HEK293T cells transfected with hTA1 and cAMP sensor, or cAMP sensor alone to control for non-specific activities. These experiments revealed that most compounds identified in the screen have very low (>10 μM $EC_{50}$) potency at hTA1

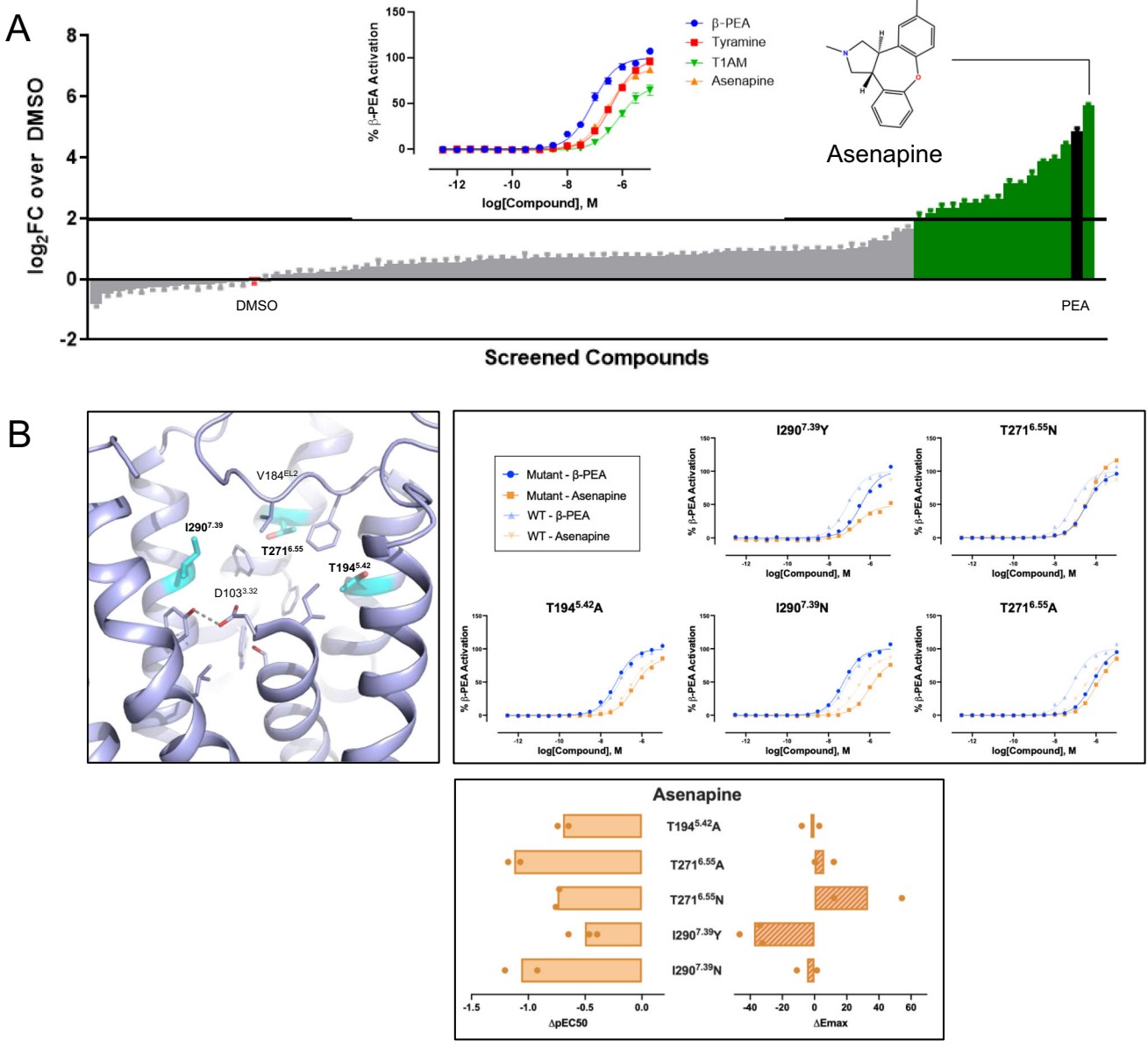

**Fig. 6 | Identification of off-target activity of aminergic neurotransmitter compounds at hTA1. A** Plot of compound activities at hTA1 at 10 uM concentration expressed as $\log_2$ fold change ($\log_2$FC) over DMSO control. The screen was performed once in quadruplicate. Data represent mean ± SEM. See Supplementary Table 3 for values. Insert shows chemical structure of asenapine, and concentration-response experiment assessing the potency and efficacy of asenapine. Data represent mean ± SEM of three independent experiments ($n = 3$). **B** SAR studies testing the effect of OBP mutations on the hTA1 activity of asenapine. Data are shown as full concentration-response curves and differences between mutant and wild type receptor with potencies determined as pEC$_{50}$s and efficacies measured as E$_{max}$. Data represent mean of two independent experiments ($n = 2$). See Supplementary Table 1 for fitted parameter values.

with the notable exception of asenapine (Supplementary Fig. S7A). Asenapine is an atypical antipsychotic, which has broadly been characterized as a potent antagonist with pan-aminergic activity at various serotonin, dopamine, histamine, and adrenergic receptors[51], though it has been reported as a partial agonist at 5-HT1AR[52]. Surprisingly, asenapine is nearly as efficacious as β-PEA (E$_{max}$=88.7% of β-PEA), exhibiting nanomolar potency (EC$_{50}$ = 273.7 nM) (Fig. 6A). Asenapine is thus a more potent agonist than the endogenous agonist T$_1$AM, with similar activity to TYR and only slightly reduced potency compared to β-PEA.

We next performed structure-activity relationship (SAR) studies to further characterize asenapine potency and efficacy, focusing on the previously highlighted residues (Fig. 6B, Supplementary Fig. 7B). Within the binding pocket, T271$^{6.55}$A reduced the potency of asenapine roughly 4-fold (EC$_{50}$ = 1152.0 nM), while T271$^{6.55}$N more marginally

reduced the drug's potency (EC$_{50}$ = 478.8 nM) while increasing its efficacy to above that of β-PEA (E$_{max}$=121.9%). Likewise, I290$^{7.39}$N reduced the potency 4-fold (EC$_{50}$ = 1001.6 nM), but without substantially impacting efficacy (E$_{max}$=84.0%). In contrast, I290$^{7.39}$Y did not impact potency (EC$_{50}$ = 275.0 nM) but substantially reduced asenapine's efficacy (E$_{max}$=51.1%). Interestingly, the T194$^{5.42}$A mutation minimally affects asenapine's potency (EC$_{50}$ = 540.3 nM), while this change otherwise dramatically changes the potencies of many of the other ligands tested (Supplementary Table 1).

Looking at mutations outside of the binding pocket, we observed that F195$^{5.43}$T slightly increases asenapine's potency by 3-fold (EC$_{50}$ = 77.1 nM). Mutations of C88$^{EL1}$S and C178$^{EL2}$S rendered asenapine a full agonist (108.0% and 102.0% of β-PEA, respectively) without impacting potency (Supplementary Fig. 7B, Supplementary Table 1).

These data show that asenapine's agonism relies on structural contacts within the OBP and thus validate that its distinct scaffold binds to the same site as previously reported hTA1 agonists. Moreover, asenapine's increase in potency when F195$^{5.43}$ is mutated to threonine is in line with its potent activity at serotonin and dopamine receptors, which all contain serines or threonines at position 5.43. Taken together, these findings further highlight the close functional relationship between hTA1 and monoaminergic neurotransmitter receptors, while uncovering key features that are unique to hTA1.

## Discussion

Herein we report the hTA1-Gs complex cryoEM structure together with functional and mutational studies. We further uncover unique receptor features that differentiate it from its fellow TAAR family members and rodent orthologs[45], and illustrate its similarity to monoaminergic neurotransmitter receptors such as 5-HT4R.

hTA1 exhibits a unique extracellular surface in which cysteine disulfide bonds connect the N-terminus to EL1 and EL2, which involves a C-($X_{6-7}$)-C motif that is observed in the N-terminal sequences of all members of the hTAAR family. Unlike mTAAR4, mTAAR5, mTAAR8c and mTAAR9, however, disrupting these bonds in hTA1 does not appear to substantially impact receptor signaling for any ligand tested herein[28]. In fact, mTA1 and rTA1 even lack the equivalent of C13$^{N\text{-term}}$, which connects to C88$^{EL1}$ in hTA1. Taken together, these data suggest that hTA1 fundamentally differs from its TAAR family members in not needing these disulfides to transduce signals, and even displays considerable structural divergence from its rodent orthologues. These findings add an additional dimension to previous pharmacological studies that uncovered surprisingly different ligand affinities at rTA1, mTA1, and hTA1[34,36]. To provide a structural context for these observations, we investigated the molecular underpinnings of these differences combining structural analysis and site-directed mutagenesis. Much of the preference for TYR in TA1 rodent orthologues appears to be attributable to an alanine in position 5.42 of rTA1 and mTA1 compared to a threonine in hTA1. This finding is counterintuitive when considering that studies of dopamine and norepinephrine receptors highlight the importance of hydrogen bonds with polar residues at 5.42 for the activity of the endogenous ligands[37,38,53,54]. This suggests that hTA1 and its rodent orthologs may have a different network of hydrogen bonding necessary for ligand-stabilized signal transduction. This is further supported by introducing a threonine at position 5.43 of hTA1 that increases TYR's potency 26-fold, whereas a S$^{5.43}$A mutation in DRD4 only lowered dopamine potency by 7-fold[37]. In contrast, the substantial change of I290$^{7.39}$ to rodent-equivalent polar residues appeared to either weaken (Y$^{7.39}$ in mTA1) or otherwise not affect (N$^{7.39}$ in rTA1) TYR potency relative to β-PEA. In the latter case, the substitution of an asparagine also substantially increased T$_1$AM potency, which, in tandem with T194$^{5.42}$A, likely contributes to the exceptionally high potency of this hormone at rTA1 relative to hTA1. Strikingly, previous studies reported that a single amino acid difference in the same position 7.39 is responsible for analogous pharmacological variation between human and rodent 5-HT1BR[55]. In the case of 5-HT1BR, several beta-blockers selectively antagonize rodent orthologs but not human 5-HT1BR, with clear implications for the study of preclinical models for both receptor pharmacology and drug development[55,56]. The herein-reported molecular insight thus highlights that the poor pharmacological and mechanistic similarity of human and rodent TA1 requires extensive validation to extrapolate findings from one species to another.

As our work uncovers surprising differences to other TAARs and particularly rodent TA1 receptors, we observe surprising similarities to neurotransmitter receptors such as serotonin receptors. Not only do we observe a similar receptor architecture compared to 5-HT4R, but the hTA1 binding pocket architecture and residue composition display key features observed in several neurotransmitter receptors. These,

for instance, include a conserved ionic bond between ligand and D$^{3.32}$, EL2 as a boundary of the pocket, and key phenylalanines in TM6 that stabilize the aromatic moiety of Ro5256390, which is likely also a feature of binding to the chemically related β-PEA. None of the other TAAR family members features both phenylalanines, though it should be noted that they do not appear to be necessary for β-PEA binding according to the recently published structure of mTAAR9[28]. Further work will thus have to be done to assess the role of these residues in ligand binding and signal transduction in hTA1.

In addition to these structural features, a screen uncovered agonist activity of several aminergic compounds at hTA1, including the potent efficacy of asenapine, a drug reported to primarily target and antagonize monoaminergic neurotransmitter receptors. While we were not surprised to find that the non-selective monoaminergic drug asenapine is able to bind hTA1 based on the structure of the receptor's orthosteric binding pocket, its potent and efficacious agonism was unexpected. Strikingly, asenapine has reported, albeit low, agonist efficacy at 5-HT1AR[52], providing pharmacological evidence for the receptor's close structural and functional relationship with serotonin receptors. These findings further underscore pharmacological overlap between hTA1 and 5-HT1AR in light of the promising clinical performance of ulotaront - a reported dual hTA1/5-HT1AR agonist. It is, however, difficult to attribute any particular dimension of asenapine's reported physiological effects to hTA1 activation, as the drug likely derives in vivo efficacy from its polypharmacology at various aminergic receptors[57,58]. Nonetheless, the reported therapeutic effects of the hTA1 agonist and antipsychotic ulotaront in clinical trials suggests that hTA1-mediated effects of asenapine could contribute to its clinical efficacy. This is particularly likely as studies showing strong brain accumulation[59] suggest that asenapine reaches sufficient concentrations to stimulate hTA1 activation in vivo. This possibility must be tempered against the uncertain future of ulotaront given its recent failure in Phase 3 trials to support its use for the treatment of schizophrenia. However, ongoing Phase 3 trials for ulotaront in the treatment of general anxiety[19] and major depressive disorders[20], as well as promising Phase 2 trials for its efficacy in treating Parkinson's disease psychosis[14], suggest that hTA1/5-HT1AR dual agonism may yet have a future in the development of novel psychiatric therapies.

Taken together, our functional and structural data provide insight into the molecular mechanisms of hTA1 and uncover critical determinants of ligand selectivity and efficacy. We further elucidate similarities and differences to fellow members of the TAAR family, rodent orthologs, and monoaminergic neurotransmitter receptors, and identify potent and efficacious hTA1 agonism by the antipsychotic asenapine. The herein presented work should thus launch investigations into asenapine's hTA1-related physiology, as well as facilitate the development of hTA1 probes with improved species- and subtype-selectivity, including those that are based on the asenapine scaffold.

## Methods

### Constructs and expression

Structural studies reported herein were performed with human hTA1 (UniProtKB Q96RJ0) modified with a F112$^{3.41}$W mutation[22] and cloned into a modified pFastBac vector. This vector included a decorated N-terminus consisting of a cleavable HA-signal sequence followed by a FLAG-tag, a 10xHis tag, a TEV protease site, BRIL, and the first 25 residues of β$_2$AR to increase expression levels. Mutations discussed herein were introduced into hTA1 using established PCR protocols (primers for each mutation are included in the Supplementary Data).

Heterotrimeric G protein was expressed from a single pDualBac virus following the previous construct design[39]. In brief, N-terminally 6xHis-tagged human Gβ1 was cloned under the control of a polyhedrin promoter, while a Gγ2-Gαs fusion construct was cloned under control of a P10 promoter. Gγ2 and Gαs were fused with a GSAGSAGSA linker. Gαs was further modified to include the mutations N271K, K274D,

R280K, T284D, I285T, G226A, and A366S to stabilize its receptor-engaged state. Nanobody 35 (Nb35)[27] was cloned into a pFastBac vector with a gp67 secretion tag. Protein expression was carried out in *Spodoptera Frugiperda* cells (*Sf*9, Expression Systems, Catalog #94-001 S, not independently authenticated) using the Bac-to-Bac Baculovirus expression system (Invitrogen), for which bacmid DNA was generated in DH10Bac cells (Invitrogen). Initial P0 virus was obtained by addition of ~3 μg recombinant bacmid DNA mixed with 3 μl FuGENE HD Transfection reagent (Promega) in 100 μl Sf900 II media (Invitrogen) to *Sf*9 cells plated in SF900 II media in wells of a 12-well plate. After 5 days at 27 °C, the supernatant was harvested as viral stock, and high-titer recombinant P1 baculovirus (>10$^9$ viral particles per ml) was generated by adding P0 to 50 ml of ~3 ×10$^6$ cells/ml and incubating cells at 27 °C for 3 days. Approximate titers were estimated by flow cytometric analysis staining P1 infected cells with gp64-PE antibody (Expression Systems, used at 1:200 dilution in 1x PBS (Gibco)). All protein for this study, including hTA1, Gs heterotrimer and Nb35 were expressed separately by infection of *Sf*9 cells at a cell density of ~2.5 ×10$^6$ cells/ml with P1 virus. After 48 hr of shaking at 27 °C, cells expressing either receptor or G protein were harvested by centrifugation at 48 h post-infection and pellets were stored at −80 °C until use. Cells expressing Nb35 were shaken for 72 hours at 27 °C, and supernatant was harvested by centrifugation and subsequent disposal of the pellet, and prepared for immediate purification described below.

## Nb35 purification

To purify Nb35, insect cell media supernatant was treated in sequence with Tris pH 8.0 (to a final concentration of 50 mM), CaCl$_2$ (final concentration of 5 mM), and CoCl$_2$ (final concentration of 1 mM), and stirred at room temperature for 1 hour to precipitate media components. Precipitate was allowed to sediment and further removed by filtration with a 0.22 μm PES Bottle Top Filter (Fisher). The final supernatant was supplemented with a final concentration of 10 mM imidazole and stirred with HisPur Ni-NTA Resin (Thermo Scientific) overnight at 4 °C. Protein-bound Ni-NTA resin was removed from supernatant by gradually removing solution from the top after sedimentation, and packed into a plastic flow column. Resin was subsequently washed with 10 column volumes (cv) of 20 mM HEPES pH 7.5, 500 mM NaCl, 10 mM imidazole, 10% glycerol. Further washing was done with 15 cv of 20 mM HEPES pH 7.5, 100 mM NaCl, 10% glycerol. Protein was eluted from the resin with 20 mM HEPES pH 7.5, 100 mM NaCl, 300 mM imidazole, 10% glycerol. The eluent was concentrated using Vivaspin 6 Centrifugal Concentrators (Sartorius). Imidazole was removed from the concentrated eluent using PD MiniTrap Sample Preparation Columns (Cytiva) according to the manufacturer's protocol. Desalted protein was concentrated, flash frozen, and stored at −80 °C.

## G protein purification

For G protein purification, insect cells were dounce homogenized in a lysis buffer consisting of 20 mM HEPES, pH 7.5, 100 mM NaCl, 1 mM MgCl$_2$, 0.01 mM guanosine diphosphate (GDP), 10% glycerol, 5 mM β-mercaptoethanol, 30 mM imidazole, 0.2% Triton X-100, and home-made protease inhibitor cocktail (500 μM AEBSF, 1 μM E-64, 1 μM Leupeptin, 150 nM Aprotinin). The cytoplasmic and membrane fractions were separated by centrifugation at 50,000 x G for 20 min. The resulting supernatant was subjected to an additional centrifugation at 200,000 x G for 45 min to further clarify supernatant. The final supernatant was bound to HisPur Ni-NTA Resin (Thermo Scientific) overnight at 4 °C. Protein-bound Ni-NTA resin was washed with 20 cv of lysis buffer lacking 0.2% Triton X-100, followed by 20 cv lysis buffer lacking 0.2% Triton X-100 and 30 mM imidazole. Protein was eluted from the resin with lysis buffer lacking Triton X-100 and supplemented with 300 mM imidazole. Eluent from the first two elution fractions after the elimination of dead volume were concentrated using Vivaspin 6 Centrifugal Concentrators (Sartorius). Imidazole was removed from the concentrated eluent using PD MiniTrap Sample Preparation Columns (Cytiva) according to the manufacturer's protocol. Eluted and desalted protein was injected onto a Superdex 200 Increase (Cytiva) size exclusion chromatography column equilibrated in 20 mM HEPES, pH 7.5, 100 mM NaCl, 1 mM MgCl$_2$, 0.01 mM guanosine diphosphate (GDP), 10% glycerol, 5 mM β-mercaptoethanol, and peak fractions containing intact heterotrimer were collected. Pooled fractions were concentrated, flash frozen, and stored at −80 °C.

## hTA1–Gs complex formation and purification

For hTA1 purification, insect cells were disrupted by thawing frozen cell pellets in a hypotonic buffer containing 10 mM HEPES pH 7.5, 10 mM MgCl$_2$, 20 mM KCl, and home-made protease inhibitor cocktail, and collected as a pellet following centrifugation at 50,000 x G. Total cellular membranes were homogenized and centrifuged twice in a high osmotic buffer containing 1 M NaCl, 10 mM HEPES pH 7.5, 10 mM MgCl$_2$, 20 mM KCl and home-made protease inhibitor cocktail. Purified membranes were directly flash-frozen in liquid nitrogen and stored at −80 °C until further use.

To form Ro5256390-bound hTA1-Gs complex, membranes were first suspended in buffer containing 10 mM HEPES pH 7.5, 10 mM MgCl$_2$, 20 mM KCl, 150 mM NaCl, home-made protease inhibitor cocktail, and 20 μM Ro5256390 (Sigma). Complexation was initiated by the addition of an excess of Gs heterotrimer and agitating at room temperature for an hour. Subsequently, apyrase (NEB) was added to a final concentration of 25 mU/mL, and the mixture was further agitated at room temperature for another hour. The mixture was transferred to 4 °C and allowed to equilibrate for 20 minutes before solubilization was initiated by the addition of a final concentration of 1% (w/v) n-dodecyl-β-D-maltopyranoside (DDM, Anatrace), 0.2% (w/v) cholesteryl hemisuccinate (CHS, Anatrace), and home-made protease inhibitor cocktail for 2 hr at 4 °C. Unsolubilized material was then removed by centrifugation at 200,000 × G for 30 min, and imidazole was added to the supernatant to a final concentration of 20 mM. Proteins were bound to TALON SuperFlow IMAC resin (Takara) overnight. Protein-bound TALON resin was washed with 15 cv of wash buffer I (25 mM HEPES, pH 7.5, 150 mM NaCl, 0.1% (w/v) DDM, 0.02% (w/v) CHS, 20 mM imidazole, 10% (v/v) glycerol, 10 μM drug). The detergent was then gradually exchanged for Lauryl Maltose Neopentyl Glycol (LMNG) by washing with 10 cv wash buffer I supplemented with 1% LMNG, followed by another 20 cv with 0.1% LMNG. Successive washes steps using 15 cv each were performed with wash buffer II (25 mM HEPES, pH 7.5, 150 mM NaCl, 0.05% (w/v) LMNG, 0.01% (w/v) CHS) and wash buffer III (20 mM HEPES, pH 7.5, 100 mM NaCl, 0.01% (w/v) LMNG, 0.002% (w/v) CHS). Complexes were eluted with 20 mM HEPES, pH 7.5, 100 mM NaCl, 0.01% (w/v) LMNG, 0.002% (w/v) CHS, 10 μM Ro5256390 and 250 mM imidazole. The eluted complex was concentrated using Vivaspin 6 Centrifugal Concentrators (Sartorius), and imidazole was removed from the protein solution by applying the sample to a PD MiniTrap Sample Column (Cytiva) according to the manufacturer protocol. Desalted complex was stored overnight at 4 °C. The next day, samples were concentrated and purified over a Superdex 200 Increase size exclusion column (Cytiva) equilibrated in 20 mM HEPES, pH 7.5, 100 mM NaCl, 0.00075%(w/v) LMNG, 0.0002% (w/v) CHS, 0.00025% GDN, and 5 μM Ro5256390. Peak fractions were pooled, concentrated to ~1.5 mg/ml, and immediately used to prepare grids for cryo-EM data collection.

## Receptor-G protein grid preparation

To prepare cryo-EM grids for imaging, 3 μl of the samples were applied to glow-discharged holey carbon EM grids (Quantifoil 300 copper mesh, R1.2/1.3) in an EM-GP2 plunge freezer (Leica). EM-GP2 chamber was set to 95% humidity at 12 °C. Sample-coated grids were blotted for

3 to 3.3 seconds before plunge-freezing into liquid ethane and stored in liquid nitrogen for data collection.

## Cryo-EM data collection and processing

All automatic data collection was performed on a FEI Titan Krios instrument equipped with a Gatan K3 direct electron detector operated by the Simons Electron Microscopy Center in the New York Structural Biology Center (New York, New York). The microscope was operated at 300 kV accelerating voltage, at a nominal magnification of 64,000x corresponding to a pixel sizes of 1.069 Å. 7,618 movies were obtained at a dose rate of 26.94 electrons per Å$^2$ per second with a defocus ranging from −0.5 to −1.8 μm. The total exposure time was 2 s and intermediate frames were recorded in 0.05 s intervals, resulting in an accumulated dose of 53.88 electrons per Å$^2$ and a total of 40 frames per micrograph.

Movies were motion-corrected using MotionCor2[60] and imported into cryoSPARC[61] for further processing. CTFs were estimated using patchCTF in cryoSPARC. An initial model was produced from a subset of micrographs using blob picking, followed by extraction, 2D classification, selection of key classes, and generation of a model ab initio. Subsequent models were produced from a curated micrograph set using particles found by template picking using the initial model. Particles were extracted, subjected to 2D classification, and a final particle stack was obtained by iterative rounds of 3D classification generating several bad models from rejected particles as a sink in subsequent heterogeneous refinement runs. A final round of 3D classification resulted in 3 separate but very similar classes that were combined to increase resolution. The composite 3-class map was further refined with NU-refinement, and the particle stack was subjected to 3D variability analysis into 3 components of 20 classes, which were examined manually. One component produced more marked changes in motion than the other two and was used to make movies in ChimeraX[62] with a volume morph function with 30 frame intermediates. The complex structure was built in Coot[63] and further refined using PHENIX[64] and ServalCat[65]. In total, we were able to confidently model residues for hTA1 in the transmembrane core and loops EL1, EL3, IL1, and IL2 (Ser19$^{N\text{-term}}$ to Ala168, Cys178$^{EL2}$ to Lys230$^{5.78}$, and Ser244$^{6.28}$ to Gly321$^{C\text{-term}}$). As discussed in the results, the C-terminus, a large portion of EL2, and the N-terminus showed considerable flexibility that prevented us from modeling individual residues, and these were represented as polyalanine chains. To attempt to further resolve the N-terminus-EL2 interface, local refinement was performed using a mask that kept hTA1 and the C-terminal helix of Gαs, and eliminated the rest of Gαs, Gβ1, Gγ2, and Nb35. This resulted in a modest increase in local resolution of portions of the transmembrane core, but did not substantially alter the resolution of the extracellular region. However, the resulting density for the more extracellular region of EL2 resembled a helical assembly, so we opted to model that portion of the polyalanine chain as a 2-turn helix, in keeping with the published structures of the mTAAR9 and 5-HT4R receptors, and the AlphaFold predicted model of hTA1. The C-terminus was extended from Helix 8 as a polyalanine chain to the end of the continuous density observed in our unsharpened maps. The density of the G protein heterotrimer enabled the modeling of residues 9-61, 205-255, and 263-394 of Gαs, residues 3-340 of Gβ$_1$, residues 6-62 of Gγ2, and residues 1-128 of Nb35. The final model was validated in PHENIX before being imported into PyMOL[66] for generating the figures shown in the manuscript.

## cAMP accumulation assays

hTA1 activity was measured via cAMP accumulation assays using the cAMP GloSensor (Promega) and essentially done as previously described[67]. Dulbecco's Modified Eagle Medium (DMEM) media (Gibco) was used for cell culture of the HEK239T cells (ATCC, Catalog #CRL-3216, not independently authenticated) used in GloSensor assays reported in this paper. DMEM with 10% v/v Fetal Bovine Serum (FBS) and 1% v/v penicillin-streptomycin (P/S) was used for regular cell maintenance and passage. Cells were incubated in a humid 37 °C incubator with 5% CO$_2$. Cells were approximately 70% confluent at the time of transfection. In preparation for transfection, 10% FBS DMEM was replaced with DMEM containing 1% v/v dialyzed FBS (dFBS) and 1% P/S. Cells were allowed to incubate for a minimum of one hour prior to transfection at 37°. Cells were then transfected with hTA1 and GloSensor DNA in a 1:1 ratio using polyethyleneimine (PEI). Wild type and mutant hTA1 were cloned into a pcDNA3 vector introducing an N-terminal HA signal sequence followed by a FLAG tag. The transfection mixture was prepared in Opti-MEM media with a ratio of 2 μL of PEI (Alfa Aesar, 1 mg/mL) per 1 μg of DNA. Transfection mixes were incubated for 20 minutes before being added dropwise to cells.

On the day following transfection, cells were plated into white, clear bottom 384-well assay plates (Greiner Bio-One) coated with poly-lysine (25 mg/ml). Cells were plated at approximately 20,000 cells per well in 40 μL of 1% dFBS DMEM media. The following day, media was exchanged for 30 μL of drug buffer (20 mM HEPES pH 7.5, 1 x Hank's Balanced Salt Solution (HBSS, Gibco), 0.1% w/v Bovine Serum Albumin (BSA) and 0.01% w/v ascorbic acid) supplemented with 1.2 mM D-Luciferin (Gold Bio). Cells were incubated in the D-Luciferin solution for a minimum of one hour at 37 °C before the addition of compounds of interest.

All drugs used in this study were dissolved into dimethyl sulfoxide (DMSO) and diluted to 1 mM stocks stored at −20 °C. These compound stocks were serially diluted in drug buffer at 3x final concentration, and 15 μL of each solution was added to each well of the assay plate.

After the addition of the compound, cells were incubated in the dark at room temperature for 30 minutes before being read in a Perkin Elmer Trilux Microbeta. Luminescent counts per second (LCPS) were reported and then plotted as a function of drug concentration and analyzed in a non-linear regression analysis of log(agonist) versus response in GraphPad Prism 8.0. Compound efficacies were normalized to the maximum activity of the endogenous agonist β-PEA. All experiments were performed in triplicate and data was averaged from two or three independent experiments as indicated. ΔpEC50 and ΔEmax values presented in the figures were calculated by subtracting the mean of a compound's pEC50 and Emax of each independent experiment from the mean of the compound's pEC50 and Emax at wild-type hTA1.

## Sequence analysis

Phylogenetic analysis, sequence alignments, and calculations of sequence similarities and sequence identities were all performed using tools of the GPCRdb[46].

## Reporting summary

Further information on research design is available in the Nature Portfolio Reporting Summary linked to this article.

# Data availability

Electrostatic potential maps and structure coordinates have been deposited in the Electron Microscopy Data Bank (EMDB) under accession code EMD-42268 (Ro5256390/hTA1-Gαs- Gβ1-Gγ2/Nb35); and the Protein Data Bank (PDB) under accession code 8UHB (Ro5256390/hTA1-Gαs- Gβ1-Gγ2/Nb35). Aligned and dose-weighted micrographs have been deposited at the Electron Microscopy Public Image Archive (EMPIAR) under accession code 11755[68]. Comparison models of the 5-HT4R-Gs complex and mTAAR9 were accessed from the PDB via accession codes 7XT8 and 8IW7, respectively. All data underlying graphs in Figs. 1, 2, 3, 5, and 6 in main text and Supplementary Figs. 1, 3, 5 and 7 are included in a supplementary source data file. Source data are provided with this paper.

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

## Acknowledgements

This work was supported by NIH grant GM133504, an Edward Mallinckrodt, Jr. Foundation Grant, a McKnight Foundation Scholars Award, an Irma T. Hirschl/Monique Weill-Caulier Trust Research Award (all to D.W.), an NIH F31 MH132317 (A.L.W), and T32 Training Grant GM062754 and DA053558 (A.L.W and G.Z.). Some of this work was performed at the National Center for cryo-EM Access and Training (NCCAT) and the Simons Electron Microscopy Center located at the New York Structural Biology Center, supported by the NIH Common Fund Transformative High Resolution Cryo-Electron Microscopy program (U24 GM129539,) and by grants from the Simons Foundation (SF349247) and NY State Assembly. We further acknowledge cryo-EM resources at the National Resource for Automated Molecular Microscopy located at the New York Structural Biology Center, supported by grants from the Simons Foundation (SF349247), NYSTAR, and the NIH National Institute of General Medical Sciences (GM103310) with additional support from Agouron Institute (F00316) and NIH (OD019994). We would like to acknowledge S. Uhl for insightful discussions in the conception of this work. This work was supported in part through the computational resources and staff expertise provided by Scientific Computing at the Icahn School of Medicine at Mount Sinai and supported by the Clinical and Translational Science Awards (CTSA) grant UL1TR004419 from the National Center for Advancing Translational Sciences. Research reported in this paper was supported by the Office of Research Infrastructure of the National Institutes of Health under award number S10OD026880 and S10OD030463. The content is solely the responsibility of the authors and does not necessarily represent the official views of the National Institutes of Health.

## Author contributions

G.Z. initiated the project. G.Z. designed experiments, expressed and purified protein, prepared grids, collected data, refined structures, performed signaling assays, and wrote the manuscript. A.K.P. performed signaling assays with help from S.Y. A.L.W. assisted with data refinement and signaling assays, and edited the manuscript. D.W. designed experiments, analyzed the data, supervised the project, and wrote the manuscript.

## Competing interests

The authors declare no competing interests.
