## [Peer Review File · Nature Communications]

Molecular Basis of Human Trace Amine-Associated Receptor 1 ActivationReviewers' Comments:

Reviewer #1:

Remarks to the Author:

Zilberg et al. report an important and interesting study of the structure of the human TA1 in complex with Gs heterotrimer. Using Cys mutational studies the work identifies hTA1 fundamentally differs from other TAAR family members in not needing disulfides to transduce signals. The study also shows hTA1 displaying significant structural differences from its rodent orthologs. Using CryoEM and mutagenesis studies their work further uncovers domains essential for ligand recognition. A range of mutagenesis studies are performed in the light of TA1-ligand complexes to ascertain the role of certain residues in ligand binding. One of the key pharmacological findings from this study showed asenapine as a more potent agonist than the endogenous agonist iodothyronamine and similar activity to other endogenous agonists. The finding that asenapine may activate HTA1 is of significant interest.

Ulotaront is a TA1 agonist and received FDA Breakthrough therapy status in the treatment of schizophrenia. This is the only TA1 agonist to reach phase III clinical trials. Why did the authors decide to not choose to resolve the structure of TA1-ulotaront complex but TA1- Ro5256390? It would be good for the authors to specifically comment on this.

The authors should also discuss their TAAR1 structure with the previously published TAAR1 models which show significant overlap in similar binding regions (see specific references listed below).

Heffernan MLR, Herman LW, Brown S, et al. Ulotaront: A TAAR1 Agonist for the Treatment of Schizophrenia. *ACS Med Chem Lett.* 2021;13(1):92-98.

Nair PC, Miners JO, McKinnon RA, et al. Binding of SEP-363856 within TAAR1 and the 5HT1A receptor: implications for the design of novel antipsychotic drugs. *Mol Psychiatry.* 2022;27(1):88-94.

The methodology reported is comprehensive and generally provides sufficient details for the work to be reproduced. However the research team did not specifically name all the 89 aminergic compounds that they tested and selected for possible TA1 agonist function and activity. It would be important to name all the 89 compounds within a supplementary table to enable the work to be reproduced.

The introduction could have mentioned the potential for TAAR 1 agonists to be used for Parkinson's L-Dopa psychosis. The authorship team may need to specifically add that recently released phase 3 trial results of ulotaront compared to placebo were unsuccessful (no significant difference found) for schizophrenia treatment.

The discussion is good and relevant.

Kindest Regards - Tarun

Professor Tarun Bastiampillai

Reviewer #2:

Remarks to the Author:

The manuscript by Zilberg et al., entitled "Molecular Basis of Human Trace Amine-Associated Receptor 1 Activation," describes the first cryo-EM structure of the trace amine-associated receptor 1 (hTA1) in complex with Gs protein and a preclinical compound, Ro5256390.

This is the first structure of the hTA1 receptor; however, the structure of another family member, mTAARf7 (TAAR9 homolog), has been deposited to PDB and pre-printed on BioRxiv in July 2023 (<https://doi.org/10.1101/2023.07.07.547762>). While both papers describe the Gs complexes of the trace amine-associated receptor family, this paper also provides a detailed assessment of the structure-activity relationship between ligand binding and species differences, and applies this knowledge to identify compounds that were not expected to target the hTA1 receptor. In other words, the existence of another paper and structure does not detract from the significance of this manuscript.

The structure of hTA1 is very interesting and has several unique and peculiar features, such as the receptor C-tail interacting with the Ga subunit, and a couple of Cys-bridges at the ECL2 that do not appear to be required for anything. Further follow-up studies (beyond the scope of the current investigation) will be needed to address these questions. This manuscript also provides a very detailed description and validation of the ligand binding site. Most importantly, the authors address the long-standing conundrum of species selectivity and the reasons why rodent models do not respond the same way to ligands identified at the human receptor. The authors also provide a detailed comparison between hTA1 and neurotransmitter receptors, which, at first glance, appeared a bit excessive; however, it elegantly led to their identification of asenapine, a pan-aminergic antagonist and a partial agonist, as a potent TA1 agonist. Overall, I think this is a great study that started with structural biology and led to the semi-rational identification of a novel target for a known drug.

While this is a great study and I do recommend this manuscript for publication, there are some questions that should be addressed:

1. Why was Ro5256390 chosen over Ro5263397? They both show similar potency in the cAMP accumulation assay.
2. The authors introduced several modifications to improve the yield of the receptor used for structural studies (F1123.41W mutation, BRIL N-terminal fusion, β -adrenergic receptor N-terminus). However, they never show that these modifications do not affect receptor activity. The F1123.41W mutation is particularly worrisome, as it was designed to improve antagonist binding at the β 2-adrenergic receptor.
3. Supp Figure 1 – please also show the phase-randomized FSC curve.
4. Supp Figure 2E – It is very confusing as to what the authors are trying to show. Related to this, the discussion about H632.44 is not particularly convincing. It is not surprising that polar residues sometimes face the lipid/micelle area. That particular histidine also makes an H-bond with the backbone of TM1, so any minor effect of its mutation can be explained by reduced receptor stability. It is also a bit perplexing why the authors focused specifically on this residue. It is true that there is some unidentified density from the micelle nearby, but the same is true in many other different areas around the receptor. It would be helpful if the authors either clarified their fascination with this residue further or removed the discussion altogether.
5. The section "Structural Similarity to Other hTAARs" is written a bit too densely for a nonspecialist audience. Perhaps simplifying this section at least a little might be helpful. It might also be helpful to emphasize why the species difference is so incredibly important for drug development.
6. While not necessary, docking of the compounds described in Fig. 2 would be very helpful to visually explain what is going on. For example, why does the R83 mutation differently affect the Ro compounds.

REVIEWER COMMENTS

Reviewer #1 (Remarks to the Author):

Zilberg et al. report an important and interesting study of the structure of the human TA1 in complex with Gs heterotrimer. Using Cys mutational studies the work identifies hTA1 fundamentally differs from other TAAR family members in not needing disulfides to transduce signals. The study also shows hTA1 displaying significant structural differences from its rodent orthologs. Using CryoEM and mutagenesis studies their work further uncovers domains essential for ligand recognition. A range of mutagenesis studies are performed in the light of TA1-ligand complexes to ascertain the role of certain residues in ligand binding. One of the key pharmacological findings from this study showed asenapine as a more potent agonist than the endogenous agonist iodothyronamine and similar activity to other endogenous agonists. The finding that asenapine may activate HTA1 is of significant interest.

We would like to thank the reviewer for their insightful comments and suggestions.

Ulotaront is a TA1 agonist and received FDA Breakthrough therapy status in the treatment of schizophrenia. This is the only TA1 agonist to reach phase III clinical trials. Why did the authors decide to not choose to resolve the structure of TA1-ulotaront complex but TA1- Ro5256390? It would be good for the authors to specifically comment on this.

We agree that it would have been more clinically relevant to determine an ulotaront-bound TA1 structure – though the compound has failed in clinical trials so far. Due to extensive competition in the structural field we instead opted to determine an hTA1 structure with the ligand that would best facilitate the formation of stable complexes, which are typically the most efficacious and potent agonists. Accordingly, we ruled out ulotaront for structural studies due to its weaker potency in cAMP accumulation assays relative to the preclinical compounds Ro5256390 and Ro5263397 (Figure 1). We additionally wanted to maximize the homogeneity of our sample, so we opted for the full agonist Ro5256390 instead of the more potent but somewhat less efficacious and smaller Ro5263397. In addition, Ro5256390 is the only agonist that retained nanomolar activity at the W264F mutation, indicating that it displays the most robust agonism among ligands and thus would be best suitable for structural work.

The authors should also discuss their TAAR1 structure with the previously published TAAR1 models which show significant overlap in similar binding regions (see specific references listed below).

Heffernan MLR, Herman LW, Brown S, et al. Ulotaront: A TAAR1 Agonist for the Treatment of Schizophrenia. ACS Med Chem Lett. 2021;13(1):92-98.

Nair PC, Miners JO, McKinnon RA, et al. Binding of SEP-363856 within TAAR1 and the 5HT1A receptor: implications for the design of novel antipsychotic drugs. Mol Psychiatry. 2022;27(1):88-94.

We apologize for the oversight and have now included a discussion of our findings in the context of these publications (line 205-209; line 326-331).

The methodology reported is comprehensive and generally provides sufficient details for the work to be reproduced. However the research team did not specifically name all the 89 aminergic compounds that they tested and selected for possible TA1 agonist function and activity. It would be important to name all the 89 compounds within a supplementary table to enable the work to be reproduced.

This is an excellent point and an oversight on our part. We have now included supplemental table 3, which lists all compounds we tested, and denotes which ones reached the 2 log₂FC threshold we assigned for further assessment.

The introduction could have mentioned the potential for TAAR 1 agonists to be used for Parkinson's L-Dopa psychosis. The authorship team may need to specifically add that recently released phase 3 trial results of ulotaront compared to placebo were unsuccessful (no significant difference found) for schizophrenia treatment.

The discussion is good and relevant.

We thank the reviewer again for this excellent suggestion and the kind words about the discussion. We have now included a reference to the promising outcome of a Phase 2 study on ulotaront's efficacy in patients with Parkinson's disease psychosis (PMID:37273942) in both the introduction (line 64/65) as well as the discussion (line 473/474). We also mention the recent failure of ulotaront in Phase 3 studies on schizophrenia in both introduction (Line 66/67) and discussion (Line 470-471), but at the same time also bring up that the compound is currently still under investigation for anxiety and major depressive disorder (Line 67/68; line 472-475). Since the recent failure of ulotaront in the 2 phase 3 studies has only been reported by the company but the clinical trials have not officially concluded, we have opted to cite clinical trials as references for these statements. The same was done for the anxiety and depression trials that are still at the recruitment stages.

Kindest Regards - Tarun

Professor Tarun Bastiampillai

Reviewer #2 (Remarks to the Author):

The manuscript by Zilberg et al., entitled "Molecular Basis of Human Trace Amine-Associated Receptor 1 Activation," describes the first cryo-EM structure of the trace amine-associated receptor 1 (hTA1) in complex with Gs protein and a preclinical compound, Ro5256390.

This is the first structure of the hTA1 receptor; however, the structure of another family member, mTAARf7 (TAAR9 homolog), has been deposited to PDB and pre-printed on BioRxiv in July 2023 (<https://doi.org/10.1101/2023.07.07.547762>). While both papers describe the Gs complexes of the trace amine-associated receptor family, this paper also provides a detailed assessment of the structure-activity relationship between ligand binding and species differences, and applies this knowledge to identify compounds that were not expected to target the hTA1 receptor. In other words, the existence of another paper and structure does not detract from the significance of this manuscript.

The structure of hTA1 is very interesting and has several unique and peculiar features, such as the receptor C-tail interacting with the Ga subunit, and a couple of Cys-bridges at the ECL2 that do not appear to be required for anything. Further follow-up studies (beyond the scope of the current investigation) will be needed to address these questions. This manuscript also provides a very detailed description and validation of the ligand binding site. Most importantly, the authors address the long-standing conundrum of species selectivity and the reasons why rodent models do not respond the same way to ligands identified at the human receptor. The authors also provide a detailed comparison between hTA1 and neurotransmitter receptors, which, at first glance, appeared a bit excessive; however, it elegantly led to their identification of asenapine, a pan-aminergic antagonist and a partial agonist, as a potent TA1 agonist. Overall, I think this is a great study that started with structural biology and led to the semi-rational identification of a novel target for a known drug.

We would like to thank the reviewer for their positive remarks on our work

While this is a great study and I do recommend this manuscript for publication, there are some questions that should be addressed:

1. Why was Ro5256390 chosen over Ro5263397? They both show similar potency in the cAMP accumulation assay.

As noted to reviewer 1, our rationale for choosing a ligand for structural studies was the use of a high-affinity full agonist. We were concerned that a partial agonist might induce additional heterogeneity in our cryoEM samples and processing, and we additionally noted that Ro5256390 had a larger scaffold with a unique ethyl substituent that would make it easier to assign the correct binding pose.

Additionally, as more and more CryoEM structures are determined and released, it has become clear that many times ligand density can be less-than-cooperative with unambiguous binding pose assignment. A readily available example is in the recently preprinted structure of the tetrabenazine-bound vesicular monoamine transporter 2 structure, where even a bulky and distinct pharmacophore does not lend itself to clear binding pose assignment from density alone

(<https://www.biorxiv.org/content/10.1101/2023.09.05.556211v1.full> Figure 3). We note that this is an outstanding area of focus in the field for technique developers, with improved ligand-modeling software continuing to be updated (PMID: 36859493).

2. The authors introduced several modifications to improve the yield of the receptor used for structural studies (F1123.41W mutation, BRIL N-terminal fusion, β -adrenergic receptor N-terminus). However, they never show that these modifications do not affect receptor activity. The F1123.41W mutation is particularly worrisome, as it was designed to improve antagonist binding at the β 2-adrenergic receptor.

This is an excellent question, and we have now included an analysis of the cryoEM construct below and in the manuscript (supplementary figure 1, supplementary table 2). The CryoEM construct displays only subtle changes when compared to wildtype hTA1 such as a minor ~ 0.5 -logfold potency increase to all compounds tested, and a relative increase to the efficacy of basically all compounds relative to β -PEA. That being said, we firmly believe that these minor differences do not affect the interpretation of our structural results. We believe the functional differences to be a result of the addition of the β 2 adrenergic receptor N-terminus, which others have demonstrated result in the localization of TA1 to the cell surface (PMID: 18524885). As for the F^{3.41}W mutation, we note that mutation of the residue at 3.41 has become an increasingly common strategy for stabilizing active state Class A GPCRs for structural determination, especially neurotransmitter receptors (For example PMID: 33762731). While the mutation W^{3.41} was initially characterized for the intention of generating inactive state crystal structures (PMID: 18222471), we note that even in this publication the authors observe activation of the β 2 adrenergic receptor mutant E^{3.41}W similar to WT by the full agonist isoproterenol.

3. Supp Figure 1 – please also show the phase-randomized FSC curve.

We have now added the phase-randomized FSC curve to Supplemental Figure 1, and are including it here for reference as well. The drop off in correlation starts at 4.52 Å, indicative that the density is not being generated from overfitting.

4. Supp Figure 2E – It is very confusing as to what the authors are trying to show. Related to this, the discussion about H63^{2,44} is not particularly convincing. It is not surprising that polar residues sometimes face the lipid/micelle area. That particular histidine also makes an H-bond with the backbone of TM1, so any minor effect of its mutation can be explained by reduced receptor stability. It is also a bit perplexing why the authors focused specifically on this residue. It is true that there is some unidentified density from the micelle nearby, but the same is true in many other different areas around the receptor. It would be helpful if the authors either clarified their fascination with this residue further or removed the discussion altogether.

We agree with the reviewer's assessment. As suggested, we have now removed the data related to the ambiguous densities of membrane components, the H63^{2,44} data, and discussion thereof as these data and findings do not substantially contribute to the overall key findings and message of the paper.

5. The section "Structural Similarity to Other hTAARs" is written a bit too densely for a nonspecialist audience. Perhaps simplifying this section at least a little might be helpful. It might also be helpful to emphasize why the species difference is so incredibly important for drug development.

We thank the reviewer for this suggestion. We have now revised this paragraph to more clearly convey that the species differences between human and rodent TA1 are unusually large, and that they have

strong implications for interpreting findings from TA1-related studies in preclinical models, and the translation of these findings into therapies. We have also rewritten the paragraph to make it more accessible to a general audience. Along these lines we have also replaced pEC50 values throughout the text with EC50 values to make potencies more accessible to readers. Similarly, we have omitted standard errors from the main text to simplify reading of sections that contain a lot of potencies and efficacies such as the one referred to by the reviewer. We instead included a sentence early on (line 89-91) stating that all values and errors can be found in supplementary table 1.

6. While not necessary, docking of the compounds described in Fig. 2 would be very helpful to visually explain what is going on. For example, why does the R83 mutation differently affect the Ro compounds.

We too would love to visualize some of the effects observed from our mutational analysis. However, we think that they might be too complex to model even with docking studies, as some of the observed effects may be due to changes in allosteric communication between receptor residues, rather than a consequence of altered ligand-receptor interactions. That is particularly true for residue R83. We believe that the presence of this residue in the periphery of the binding pocket likely limits its role in coordination of ligand binding. To this end, we don't believe that docking would help resolve the precise role that R83 and mutations thereof would play in.

Reviewers' Comments:

Reviewer #1:

Remarks to the Author:

The authors have addressed all my comments satisfactorily.

I have no further comments. The work is of significance to the field and the methodology is sound.

There is sufficient detail for the work to be reproduced.

Reviewer #2:

Remarks to the Author:

I thank authors for answering my questions, I think the manuscript reads much better.

However, I do have an outstanding concern about the map (s).

As per my previous suggestion, authors provided a phase-randomised FSC curve, and in their rebuttal they state that "The drop off in correlation starts at 4.52 Å, indicative that the density is not being generated from overfitting."

However, this curve looks suspicious. The ideal phase-randomized curves for a high quality SPA model will exhibit a sharp drop of to 0 (or almost 0) at the low resolution (usually the low-pass resolution of the starting map for refinement) and then lie flat. This is not what's happening in this case. The drop off happens quite late (authors point out that this is the 4.52Å) and then it never goes to 0.

The underlying cause for this could be due to a number of factors:

1. Potential overfitting, and by this it could be due to particles that have been included in the final 3D refinement which are either contaminants or particle projections which are structurally heterogeneous relative to the consensus refinement.
2. The poor alignment of at least some particles, as the authors used cryoSPARC, careful inspection of the posterior precision plots could be informative
3. If the 3D mask used for final 3D refinement is too 'tight' around the observed density, leading to Fourier edge artifacts which lead to noise correlation at higher spatial frequencies.

It is my opinion that this issue requires attention. In addition, no map-model FSC curves were provided, this would also somewhat address any concerns around the quality and interpretation of the SPA analysis.

In my opinion the issues with the map and the curves need to be addressed prior to publication.

Reviewer #1 (Remarks to the Author):

The authors have addressed all my comments satisfactorily.

I have no further comments. The work is of significance to the field and the methodology is sound. There is sufficient detail for the work to be reproduced.

We appreciate the comments and helpful suggestions of the reviewer, and thank them for taking the time to look over our manuscript.

Reviewer #2 (Remarks to the Author):

I thank authors for answering my questions, I think the manuscript reads much better.

However, I do have an outstanding concern about the map (s).

As per my previous suggestion, authors provided a phase-randomised FSC curve, and in their rebuttal they state that ???The drop off in correlation starts at 4.52 ??, indicative that the density is not being generated from overfitting.???

However, this curve looks suspicious. The ideal phase-randomized curves for a high quality SPA model will exhibit a sharp drop of to 0 (or almost 0) at the low resolution (usually the low-pass resolution of the starting map for refinement) and then lie flat. This is not what???'s happening in this case. The drop off happens quite late (authors point out that this is the 4.52???) and then it never goes to 0.

The underlying cause for this could be due to a number of factors:

1. Potential overfitting, and by this it could be due to particles that have been included in the final 3D refinement which are either contaminants or particle projections which are structurally heterogeneous relative to the consensus refinement.
2. The poor alignment of at least some particles, as the authors used cryoSPARC, careful inspection of the posterior precision plots could be informative
3. If the 3D mask used for final 3D refinement is too 'tight' around the observed density, leading to Fourier edge artifacts which lead to noise correlation at higher spatial frequencies.

It is my opinion that this issue requires attention. In addition, no map-model FSC curves were provided, this would also somewhat address any concerns around the quality and interpretation of the SPA analysis. In my opinion the issues with the map and the curves need to be addressed prior to publication.

We thank the reviewer for their helpful comments and valid concerns regarding the tailing drop-off of the phase-randomized curve presented in the last revisions. We revisited the particle stack used to generate our model, and examined the possibilities offered for signs of which might be contributing to the phase-randomized curve not quite reaching 0 before the nominal gold-standard FSC curve reached 0.143.

To test if the tightness of the mask is the culprit for the odd non-zero tail of the phase-randomized correlation, we reran non-uniform refinement using different distances for the dynamic mask edge. The default values were 6A for the near value and 14 for the far, so we opted to increase both values to loosen the mask. This led to a negligible decrease in the value of the phase-randomized curve at the 0.143 mark for the tight-mask FSC, but not to 0 by any means.

Dynamic mask threshold (0-1)	<input type="text" value="0.2"/>	Level set threshold for selecting regions that are included in the dynamic mask. Probably don't need to change this
Dynamic mask near (A)	<input type="text" value="10"/>	Controls extent to which mask is expanded. At the near distance, the mask value is 1.0 (in A)
Dynamic mask far (A)	<input type="text" value="18"/>	Controls extent to which mask is expanded. At the far distance the mask value becomes 0.0 (in A)

Further, inspection of the posterior precision plot of the original particle stack did not reveal any particularly concerning asymmetries or preferred orientation of particles.

Before proceeding with further in-depth investigation of what might be wrong with the particle stack, we opted to calculate a map-model FSC using Phenix's mtriage program. This yielded curves that suggest a map-model FSC resolution of ~ 3.65 (unmasked) and ~ 3.30 (masked) at 0.143.

Lastly, we examined the particle stack used to generate our model, and ran a 2D classification to see if there were any egregiously bad particles that flew under the radar of our initial 2D classification. In doing so, we found that roughly ~11% of particles formed 2D classes that did not appear to resemble the TA1-Gs complex, and so we discarded them and re-ran the non-uniform refinement for our model. This provided a map that appears globally near-identical to our original map, with the same global resolution of 3.37Å global resolution. Examining the phase-randomized curve, we now observe a full drop to 0.

Subsequent local refinement of the receptor alone appeared to substantially improve the quality of the phase-randomized FSC curve, without impacting global resolution.

As for the edge in the drop off of the curve. Part of that is an inherent and well known artifact in cryoSPARC as previously discussed in a forum thread by cryoSPARC staff and Professor John Rubinstein:

<https://discuss.cryosparc.com/t/tight-corrected-and-loose-gsfsc-curves/201/3>

JohnRubinstein

Mar '17

Hi All,

The dip is an artifact of the boundary where phases are randomization and where they are experimental. The same artifact is seen in Relion's phase randomization for mask correction and extensive discussions about it can be seen in ccpem's archive. It has been suggested that the 'dip' points be removed from the curve to make it look nicer, but I believe it is preferable to leave them in to show that correction has been done. If the corrected FSC is lower than the tight masked FSC then the correction is doing its job: it is showing that some of the correlation in the tight masked FSC is due to masking artifacts. The corrected FSC should always be used to interpret resolution.

I will leave it to @apunjani to comment on the details of the automasking approach that he implemented.

Best regards,
John

Lastly, we still observe tailing of the phase-randomized curve from ~4.1Å onwards, which we attribute to the conformational heterogeneity observed within the complex. This conformational heterogeneity has been demonstrated by the supplementary movies we supplied showing the motion of the transmembrane helices relative to the heterotrimeric G protein.

In addressing the reviewers concerns, we have now accordingly updated the densities in Supplementary Figure 2 using the map with 11% of particles removed. Additionally, we have updated the flow-chart for our methodology to include the 2D classification at the end, and updated the FSC curves and local resolution map. Lastly, we have deposited the pertinent micrographs used to determine this structure into EMPIAR (EMPIAR-11755) for public examination. This allows interested readers to reproduce the data herein presented, in the interest of full transparency.

We would like to point out that this additional work had no effect on the density for the compound, the ligand binding pocket residues, and overall model, and therefore does not affect the overall message and findings of the paper.

Reviewers' Comments:

Reviewer #2:

Remarks to the Author:

I thank authors for a very detailed response for my previous comments.

All of my concerns about the quality of the maps have been alleviated.

This is a great study and I am looking forward to seeing it published.